# The Identification of Dual T-Cell and B-Cell Epitopes Within Viral Proteins Utilizing a Comprehensive Peptide Array Approach

**DOI:** 10.3390/vaccines13030239

**Published:** 2025-02-26

**Authors:** Binghang Liu, Muqun Bai, Fei Zheng, Mingchen Yan, Enen Huang, Jie Wen, Yingrui Li, Jun Wang

**Affiliations:** 1iCarbonX, Zhuhai 519000, China; liubinghang@icarbonx.com (B.L.); baimuqun@163.com (M.B.); wenjie@163.com (J.W.); 2Shenzhen Digital Life Institute, Shenzhen 518000, China; 3Faculty of Health and Medical Sciences, University of Surrey, Guildford GU2 7YH, UK; 4State Key Laboratory of Quality Research in Chinese Medicine, Macau University of Science and Technology, Taipa, Macau, China

**Keywords:** epitope, immunoreactivity, porcine parvovirus, PPV, peptide vaccine, peptide array

## Abstract

**Background/Objectives:** The development of vaccines that elicit both T-cell and B-cell responses is crucial for effective immunity against pathogens. This study introduces a novel approach to identify precise epitope peptides within viral proteins that can stimulate both arms of the adaptive immune response, using Porcine Parvovirus (PPV) as a model. **Methods:** Mice were infected with PPV, and a peptide array was utilized to detect IgG signals in their sera. This approach facilitated the assessment of the immunogenicity of the PPV proteome, leading to the identification of 14 potential epitope candidates. These candidates were then used to immunize additional mice, and their ability to induce T-cell and B-cell responses was evaluated. **Results:** The immunization experiments identified an optimal peptide, P6, which robustly activated both T cells and B cells. Further analysis of the sub-regions of this peptide confirmed P6 as the most potent inducer of immune responses. The anticipated epitope was detected in mice immunized with P6, highlighting the efficacy of our method in identifying epitopes that engage both T cells and B cells. **Conclusions:** This study presents a novel strategy for the identification of dual T-cell and B-cell epitopes by directly evaluating the immunoreactivity of antibodies in serum. This finding holds significant promise for the advancement of epitope-based vaccines.

## 1. Introduction

Vaccine-induced responses typically involve B-cell-mediated antibody responses and T-cell responses [1]. Conventional vaccination strategies, including the use of inactivated or attenuated infectious agents, as well as epitope-based approaches, such as protein, DNA, and RNA vaccines, rely on their ability to elicit both B-cell antibody responses and T-cell reactions to ensure protection against pathogens [1,2,3,4,5]. An immunogen containing epitopes capable of stimulating both T-cell and B-cell responses is theoretically the most efficient. However, such dual-specificity immunogens also become prime targets for immune evasion strategies employed by viruses [6], highlighting the critical need for research into dual T-cell and B-cell epitopes.

While T-cell epitopes have been relatively well-characterized [1,7,8] and can be identified using a variety of in silico approaches [5,9,10,11,12], identifying B-cell epitopes presents a significant challenge due to their conformational nature [10,13]. This inherent complexity reduces the likelihood of a short natural linear peptide possessing both T-cell and B-cell epitopes. Recent advancements in deep learning methods have made it possible to design peptides or proteins that mimic the structural features of B-cell epitopes [14,15,16], thereby facilitating the generation of peptides that cover dual T-cell and B-cell epitopes. However, precisely verifying their immunoreactivity through animal immunization experiments remains a significant bottleneck in this field.

Peptide arrays have emerged as a promising tool for assessing epitopes by evaluating serum immunoreactivity [17,18]. These arrays are characterized by their industrial-grade stability and decreasing costs. They can be broadly categorized into two types: (1) comprehensive sets of short peptides derived from a virus’s proteome [19,20,21], which are more commonly accepted by biologists but are limited by the constraints of the source proteins and linear epitopes, and (2) arrays of random peptides akin to phage display libraries [22]. The first type has been widely used in early COVID-19 pandemic studies, with many tests developed using linear peptides from SARS-CoV-2 [19,21,23]. However, these microarrays often failed to capture strong signals, particularly in the immunodominant Receptor Binding Domain (RBD) of the Spike protein [24], leading to the general consensus that this type of microarray is unsuitable for evaluating conformational epitopes. In contrast, the second type of peptide array offers a spectrum of signal intensities ranging from weak to strong, which aligns well with the training requirements of deep learning models. Theoretically, this approach allows for the exploration of a broader range of immunoreactive peptides and shows promise in evaluating conformational regions. To validate the efficacy and significance of this model, we have designed a study to investigate its potential.

In this study, we employed a peptide array-based approach to identify T-cell and B-cell epitopes within the Caspid1 protein of Porcine Parvovirus (PPV), a significant pathogen affecting swine reproduction [25]. Firstly, we infected mice with PPV and utilized peptide arrays to detect IgG signals in their sera, thereby identifying potential epitopes. The identified epitopes were then synthesized and used to immunize additional mice, enabling us to evaluate their immunogenicity. Finally, our results revealed a single peptide that functioned as both a T-cell and B-cell epitope, eliciting immune responses in immunized mice. As confirmed by a peptide array assay, antibodies produced by peptide-immunized mice exhibited reactivity against this dual-functional epitope.

## 2. Materials and Methods

### 2.1. PPV and Peptide Synthesis

In this study, PPVA30 was procured from the China Institute of Veterinary Drug Control, located in Beijing, China, and subsequently cultivated in Swine Testicular (ST) cell line (ZhongAn Biological Pharmaceutical Co., LTD, Changsha, China) with RPMI1640 medium. The peptides utilized to immunize mice were synthesized by GL Bio-Chem (Shanghai) Ltd., Shanghai, China, with a guaranteed purity of at least 90%. All the peptide sequences were included in Appendix A.

### 2.2. Mice and Study Designs

Specific pathogen-free (SPF) female BALB/c mice (six-eight weeks old) were purchased from Charles River. All the experimental mice procedures complied with the relevant guidelines and regulations for the management and welfare of experimental animals approved by the Ethics Committee of China Technology Industry Holdings (Shenzhen, China) Co., Ltd. (license number: SYXK(Yue)-2021-0263).

Animals (*n* = 5 or 6 per group) were subcutaneously immunized at four sites in the armpits of the front and rear limbs with either a virus (VI) or peptides (PIs). In the VI group, PPV virus, purified by sucrose density ultracentrifugation, was utilized, whereas in the PI group, 20 μg/peptide/mouse was administered in conjunction with Montantide ISA 601AG and CpG ODN, Class B (murine)—TLR9 agonist. Mice were fasted from water for 6 h prior to immunization. Immunization was conducted on days 1, 21, and 42. Serum samples were collected on days 28 and 49, which were further detected with ELISA or peptide array. Spleen tissues were isolated from mice on day 28 (for both VI and PI groups) or day 49 (only for the VI group). These samples were further detected with ELISpot assay.

### 2.3. Enzyme-Linked Immunosorbent Assays (ELISA)

The binding of PPV or peptide-specific antibodies was evaluated using an ELISA assay. Initially, 96-well plates were coated with 10 μg/mL of peptides or 5 μg/mL of virus in FBS-PBST and incubated at 4 °C overnight. On the following day, plates were washed with wash buffer (0.05% Tween-20 in FBS-PBST) and then blocked with 350 μL of casein block per well at room temperature. After 2–3 h, the block solution was discarded, and plates were blotted dry. Then, two-fold serial dilutions of serum in casein block were added to the wells, followed by incubation for 1 h at room temperature. The plates were then washed five times and incubated for 1 h with 0.1 μg/mL of Anti-Mouse IgG-HRP antibody (Invitrogen, Waltham, MA, USA) in casein block, also at room temperature and in the dark. After another five washes, 100 μL of TMB/H_2_O_2_ was added to each well. The reaction was stopped by adding 50 μL of H_2_SO_4_ per well. The absorbance at 450 nm was measured using an ELISA plate reader (Multiskan Sky, Thermo, Waltham, MA, USA). The endpoint titers were determined as the highest reciprocal dilution of serum that resulted in an absorbance value exceeding 1.8 times the background level.

### 2.4. Enzyme-Linked Immunospot (ELIS Pot) Assay

To detect antigen-specific T-lymphocyte responses, an IFN-γ-based ELISpot assay was performed. Briefly, spleens were removed from mice and splenocytes were isolated. Following the instructions in Mouse IFN-γ ELISpot kit (Mabtech, 3321-4AST-2, Stockholm, Sweden), plates were washed four times with sterilized PBS, 200 μL/well. A total of 50 μL lymphocyte culture (5 × 10^6^ cell/mL) was added to the reaction well and left at room temperature for at least 30 min. A total of 50 μL (200 μg/mL) of peptides was then added to each well and incubated at 37 °C with 5% CO_2_ for 40 h. Negative controls were defined as cells incubated without stimulation. After washing five times with PBS, plates were processed in turn with biotinylated IFNγ detection antibody (R4-6Ab-Biotin, 1 μg/mL), Streptavidin–ALP conjugate, and BCIP/NBT-plus substrate. Finally, development was halted by rinsing with deionized water when the colored spots were intense enough to be visually observed. The numbers of spots were then calculated with Elispot (AID, Germany) reader software V7.0. INF-γ^+^ T-cell counts per 2.5 × 10^5^ lymphocytes were used for data analysis.

### 2.5. Antibody Binding Signals Detected with High-Density Peptide Microarray

The manufacturing of the peptide microarray used in this work has been described previously [17]. The library of 131,712 random sequences was synthesized by using 16 of the 20 natural amino acids (Cys, Met, Ile, and Thr excluded). Each peptide includes 5–13 amino acids in length that contain 99.9% of all possible arrangements of 4-mer peptides and 48.3% of all possible arrangements of 5-mer peptides. A common linker sequence, GSC, was synthesized on the amine terminus of all peptides.

Peptide microarrays were pre-washed in ultrapure water for 30 min and then exposed to a 1:100 dilution of serum for 1 h at 37 °C. After that, 5 nM of Alexafluor 555-conjugated anti-human secondary antibody was added to detect the peptide-bound antibodies in 0.75% casein at 37 °C. One hour later, arrays were washed to remove any unbound material and scanned in a Perkin Elmer Scan Array laser scanner at 560 nm and 10 µm resolution. The generated tiff images were aligned to the corresponding .gal file. A .gpr file was produced containing the intensity values corresponding to the appropriate peptide.

### 2.6. Immunoreactivity Model with Antibody Binding Signals

For each sample, the peptide sequences were first encoded using one-hot encoding. The peptides and their corresponding signal intensities were then randomly divided into three datasets: a training set (60%), a validation set (20%), and a testing set (20%). A deep learning framework was subsequently developed to construct a regression model using the training and validation sets. The architecture of this model is illustrated in Appendix A. The performance of the regression model was evaluated by calculating the correlation between the detected and predicted values from the testing set. Since the peptide signals reflect the binding ability of peptides with antibodies in serum, this regression model effectively evaluates the antibody reactivity of the peptides. We defined the predicted value from the model as the immunoreactivity score. To determine the immunoreactivity score for each site within the protein sequence, we generated all possible 13-amino-acid peptides derived from the sequence and used the regression model to predict their scores. The immunoreactivity score for a specific site was calculated as the maximum score among all peptides that encompassed that site.

### 2.7. Sequence and Known Epitope Data Collection

The amino acid sequences for the capsid protein 1 (Protein accession ID: NP_757371.1, Uniprot ID: P52501) of Porcine Parvovirus (GCF_004132625.1, ASM413262v1) were obtained from NCBI database. 48 known epitopes were collected from IEDB database with Uniprot ID P52501. All these information was included in Appendix A.

### 2.8. MHC Binding Prediction and Antigenicity Prediction

netMHCpan-4.0 from IEDB were used to predict the MHC binding of 9 to 14-mer peptide sequences [7]. Site MHC affinity was defined by the maximum predicted affinity of peptides covering it. Antibody Epitope Prediction Standalone version 3.1 from IEDB was used to predict the Kolaskar and Tongaonkar antigenicity [26] of each site. The predicted MHC affinity and antigenicity of each site were included in Appendix A.

### 2.9. Quantification and Statistical Analysis

Statistical tests were performed in Prism8 or Python using scipy and statsmodels packages. Graphics were plotted with Prism 8 or Python with seaborn package. Both statistical test and graphics were analyzed using Python 3.9.16. Depending on the data, Mann–Whitney U, ANOVA with Tukey HSD post-hoc, and Student’s *t*-test comparison tests were used for statistical analyses, as appropriate. In all analyses where statistical significance was tested, significance was defined as: *** *p*-value < 0.001; ** *p*-value < 0.01; * *p*-value ≤ 0.05.

## 3. Results

### 3.1. Overview of the Strategy

The methodological flowchart outlining the approach to identify B- and T-cell epitopes for vaccine peptide design is depicted in Figure 1, which encompasses three key stages: (i) Discovery, where candidate epitopes are identified; (ii) Selection, where optimal candidate peptides are selected; and (iii) Precision, where precise epitopes and optimal peptides are determined.

During the Discovery stage, infected serum samples or serum samples from immunized models are screened using a peptide array to identify epitope regions of the proteome that might elicit robust antibody responses. Concurrently, in silico MHC binding predictions are employed to select epitopes for eliciting T-cell responses.

In the Selection stage, peptides encompassing immunoreactive epitopes and exhibiting a high MHC binding affinity are synthesized and used for animal immunization. These peptides vary in length, with their number determined by experimental throughput. Antibody titers and T-cell responses are assessed in immunized animals to select those peptides inducing strong immune reactions for further study.

Finally, in the Precision stage, selected peptides are further fragmented and used to immunize additional animals. Beyond monitoring B- and T-cell responses, peptide arrays are utilized to characterize epitopes in immunized samples, confirming their consistency with the immunized peptides.

In this research, this approach involved infecting mice with PPV, followed by screening serum immune reactions using peptide arrays, ultimately leading to the identification of precise T and B shared-epitope peptides.

### 3.2. Discovery of T and B Shared Epitopes in PPV Proteins

Fifteen mice were inoculated with the PPV virus, and serum samples were collected post-inoculation to measure their PPV-specific antibody titers (Appendix A). These samples, along with six more from negative controls, were detected with a peptide array, resulting in a total of 21 samples. This detection allowed us to develop an immunoreactivity model for each sample based on the peptide signal data. Following this, we applied filtering criteria to exclude samples with weak immune responses, ultimately retaining five negative and ten positive samples (Appendix A). These models were then used to predict the immunoreactivity of all possible 13-amino-acid peptides within the Capsid1 protein of PPV, which has been identified as the immune-dominant protein [27]. The immunoreactivity score for each site was calculated for every sample.

In parallel, we predicted the MHC binding affinity of each peptide (9–14 amino acids in length) within the Capsid1 protein using the netMHCIIpan algorithm [7]. Subsequently, the MHC affinity at each site was determined. By integrating both the immunoreactivity scores and MHC binding affinities, we identified and synthesized 14 peptides for further investigation (Figure 2, Appendix A).

### 3.3. Selection of Peptides Inducing B- and T-Cell Immune Responses

Ten additional mice were immunized with a cocktail of all peptides, as outlined in Figure 3A. Sera and splenic lymphocytes were harvested from the mice on Days 28 and 49 post-immunizations. They were further used to evaluate the antibody responses and T-lymphocyte responses, respectively.

The PPV-specific antibody titers in the sera were measured using ELISA plates coated with purified PPV virus. The titers ranged around 1:1024 around Day 28 and increased up to 1:4096 at Day 49 (Figure 3B). The significant increase in PPV antibodies after Boost2 indicates a robust production of PPV-specific IgG antibodies following immunization. The peptide-coated ELISA results also support this finding (Figure 3C). After Boost1, significant peptide-specific IgG antibody responses were elicited for peptides P5, P6, P8, P9, and P12 (Figure 3E). All the mice had positive responses to P6, P9, and P12 (Appendix A). With Boost2, the specific IgG antibody responses against peptides P5, P6, P10, P11, P12, and P14 were further significantly enhanced (Figure 3F).

In vitro, splenic lymphocytes from immunized mice were stimulated with eleven distinct antigenic peptides. Peptide-specific splenic IFN-γ-secreting T lymphocytes were quantified using ELISpot assays. Figure 3D illustrates that both two-dose and three-dose immunizations elicited significant peptide-specific splenic IFN-γ-secreting T-lymphocyte responses when compared to the adjuvant control group. Unlike the continuous increase in IgG titers, the peak response was observed after the two-dose immunization. In the ELISpot assays after Boost1 (Figure 3G and Appendix A), among all the tested peptides, we found that P6 and P14 induced the most robust peptide-specific IFN-γ-secreting T-lymphocyte responses.

These findings indicate that the mixed peptides can effectively induce substantial B-cell and T-cell responses, especially responses to P6, P11, and P14. Among the peptides, P6 induced the highest T-cell response, achieved the highest antibody titer at Boost1, and its specific antibodies could be detected in all immunized mice even after the second immunization. Thus, this single antigenic peptide P6 could induce a significant P6 peptide-specific IgG antibody response and IFN-γ-secreting T-cell immune response.

### 3.4. Validation and Precision of P6 Vaccine Peptide

Given that the length of the P6 peptide is up to 50 amino acids (Appendix A), we derived six sub-peptides based on their immunoreactivity score as depicted in Figure 2. These sub-peptides were administered to ten mice for immunization, following the strategy outlined in Figure 3A. For comparison, an additional ten mice were immunized with the full-length P6 peptide. The peptide-specific IFN-γ-secreting T-lymphocyte responses were assessed using Boost1 samples. Boost2 serum samples were analyzed using a peptide-coated ELISA.

Sub-peptide P6-3, similar to the full-length P6, exhibited heightened IFN-γ-secreting T-lymphocyte responses (Figure 4A,B), while P6-3 also induced higher antibody levels than the other peptides (Figure 4C). However, the mixture of shorter peptides induced significantly lower IgG titers compared to the whole P6 peptide (*p* = 0.011). The elevated IgG responses to P6-2 and P6-4 in some mice immunized with P6 (Figure 4B) indicated that the B-cell epitopes in the P6 peptide that are responsible for inducing IgG do not solely reside within P6-3.

To further make clear the precise epitopes in the P6 peptide, we immunized six more mice with P6 and employed a peptide array assay. After training the immunoreactivity model for each sample (Appendix A), we found that three mice had strong P6-3 immunoreactivity, while two mice had strong P6-4 immunoreactivity and the control mice did not show immunoreactivity to P6 (Appendix A). The predicted immunoreactivity scores of each sample revealed a significant high reactivity region, which encompassed the full P6-3 and portions of P6-4 peptide (Figure 4D). These results matched well with the IgG titers above and are also supported by the predicted antigenicity using in silico software (Appendix A).

However, we also noticed that in PPV-infected mice at the beginning, both the P6-3 and P6-4 regions have low immunoreactivity, similar to the control mice (Figure 4E). This might mean that the P6-stimulated antibodies might be different to the PPV-evoked antibodies.

## 4. Discussion

This study presents an innovative methodology for the precise identification of epitopes capable of simultaneously eliciting both B-cell and T-cell responses. Through the integration of peptide array experimental data and advanced bioinformatics analyses, we successfully identified peptides that induce robust antibody responses while also eliciting peptide-specific IFN-γ-secreting T-lymphocyte responses.

The peptide P6, which induced the highest responses, corresponds to the known epitope 104,369 in the Immune Epitope Database (IEDB) [28], thereby validating our findings. Furthermore, our results are consistent with the IEDB predictions, including both MHC affinity (Figure 2) and B-cell antigenicity (Appendix A), both of which indicate a peak value in the p6-3 region. All of the predicted results and peptides utilized in this study are included in the Appendix A. These findings hold promise for advancing the development of PPV vaccines aimed at reducing reproductive failure in swine [29]. Notably, it is important to acknowledge that the MHC affinity predictions were based on murine MHC, which may require recalibration using swine MHC for optimal relevance.

In this study, we observed discrepancies in immunoreactivity between the peptide-immunized and PPV-immunized mice (Figure 4D,E), potentially reflecting differences in their simulated antibody responses. These differences could be attributed to the P14 and P11 peptides, which may also elicit dual B- and T-cell responses. While neutralization experiments would provide further insight into the biological efficacy of antibodies simulated by the P6 peptide, this immunoreactivity discrepancy poses a fundamental challenge in designing efficient peptide vaccines. To address this issue, we propose an approach involving the generation of peptides through scaffolding functional epitope sites [14]. This strategy is grounded in the structural characterization of target antigens from infectious agents and immunodominant epitopes of known neutralizing antibodies. Recent advancements in AlphaFold protein structure predictors [30] and peptide array technologies have made it feasible to obtain these two critical pieces of information at a relatively low cost. Moreover, incorporating peptide array assays during the selection stage would be essential for validating the epitope consistency of artificially designed peptides.

The effectiveness of our approach has been demonstrated through its successful application in identifying linear epitopes across the entire PPV antigen. However, one limitation of this study is that only linear peptides were evaluated and synthesized for immunization in mice. Nevertheless, the developed immunoreactivity models may also prove valuable for assessing conformational epitopes, as they can evaluate peptide reactivity regardless of its source. Recent findings have indicated that these models can assess the reactivity of local regions on the surface of antigen proteins ([31], manuscript in preparation), further expanding their potential applications.

The scope of our strategy extends beyond the identification of natural linear peptides that can elicit both B- and T-cell responses. In the Discovery stage, incorporating samples from recently convalescent patients or those with confirmed neutralizing effects enables our method to identify the immunodominant epitopes associated with neutralizing antibodies. Furthermore, peptides that mimic the structure of these immunodominant epitopes in antigens can be generated after the Discovery stage. These peptides could be linked to known T-cell epitope peptides to form artificial multiple epitope peptides, which can elicit both B- and T-cell responses [2]. Recent studies have demonstrated the efficacy of polypeptide vaccines that link multiple epitopes in inducing robust immune responses [32]. Additionally, in the Precision stage, we can evaluate whether the chosen peptides can simulate antibodies targeting specific regions. Future research will focus on exploring a broader range of natural and artificial peptides.

The applicability of our strategy is not limited to prophylactic vaccines, which typically aim to induce long-term B-cell and T-cell memory. Our approach can also be leveraged to develop therapeutic vaccines, offering a new avenue for treatment. While current therapeutic vaccines often focus on activating or enhancing cellular immunity [33,34], artificial peptides that cover tumor-specific antigen (TSA) epitopes and tumor-associated antigen (TAA) epitopes [35] may hold promise in eliciting both T-cell and B-cell responses. This could provide a more comprehensive immune response, targeting multiple aspects of the disease. By exploring the potential of our strategy in therapeutic vaccine development, we can unlock new possibilities for treating diseases and improving patient outcomes.

## 5. Conclusions

In conclusion, our study demonstrates the successful identification of three immunogenic peptides on the Caspid1 protein of PPV, which can elicit both T-cell and B-cell responses. Notably, we refined our search to identify a sub-peptide with a length of just 14 amino acids, highlighting the potential for precise epitope discovery. This approach represents a methodological advancement in epitope identification, especially in direct evaluation of antibody immunoreactivity in serum.

Furthermore, our observation that peptide-induced epitopes may not always match those induced by the virus itself underscores the importance of considering spatial epitopes in vaccine design. While our peptide microarray-based immunoreactivity assessment and epitope discovery strategy are designed to address this challenge, further experimental validation and project support are necessary to fully demonstrate their feasibility.

Overall, this study contributes to the development of more effective vaccines by providing a framework for precise epitope identification and highlighting the need for spatial epitope consideration in vaccine design.

## Figures and Tables

**Figure 1 vaccines-13-00239-f001:**
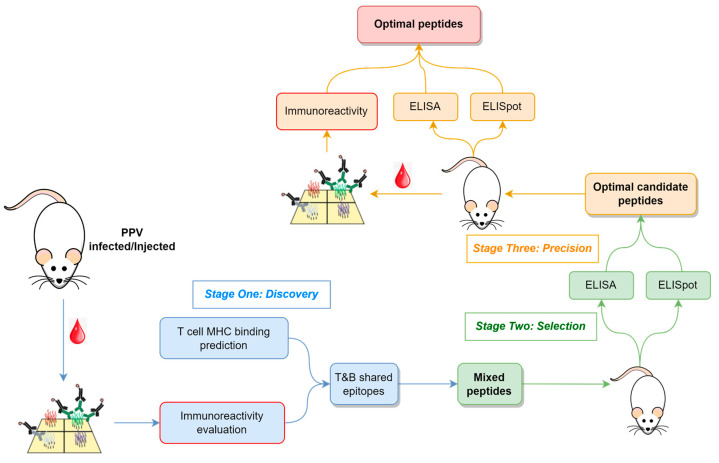
Overview of the strategy to find T- and B-cell epitope peptides. The whole pipeline includes three main stages: Discovery, Selection, and Precision. Immunoreactivity was evaluated using a peptide array in the first and last stages (red border).

**Figure 2 vaccines-13-00239-f002:**
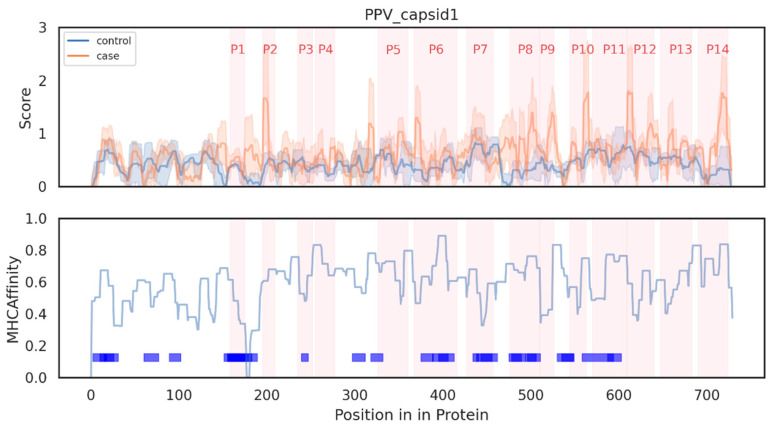
Epitopes in Capsid1 protein of PPV. Scores were calculated by immunoreactivity models. MHC affinities were predicted by NetMHCIIpan and only affinities to H-2-Db were shown here. Selected peptides were shown in red regions. Blue rectangles stand for known PPV epitopes in IEDB.

**Figure 3 vaccines-13-00239-f003:**
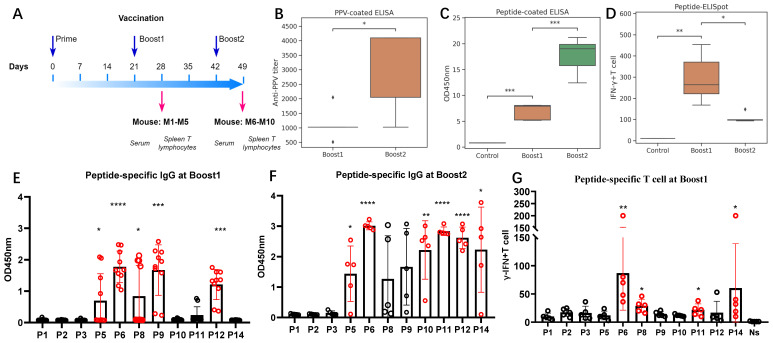
Immunogenicity evaluation of mixed peptides vaccination. (**A**) Groups of BALB/c mice (*n* = 5) were immunized with three injections of mixed peptides or with placebo via i.m. route. Sera were collected at 4- and 7-weeks post immunization. (**B**) Anti-PPV IgG titers were measured for each mouse after Boost1 and Boost2. (**C**) Peptide-specific IgG responses were assessed by peptide-ELISA. Total OD450 nm values (sum of individual peptide values) are shown for each boost, demonstrating an increasing trend with successive boosts. (**D**) ELISpot assay was performed to evaluate the capacity of splenocytes to secrete IFN-γ following re-stimulation with PPV peptides. *Y*-axis shows INF-γ^+^ T-cell count per 2.5 × 10^5^ lymphocytes. (**E**) Peptide-specific antibody detected after Boost1. P* in the *X*-axis stands for peptide. (**F**) Peptide-specific antibody detected after Boost2. P* in *X*-axis stands for peptide. (**G**) IFN-γ-secreting T-lymphocyte responses to peptides for each mouse after Boost1. P* in *X*-axis stands for peptide and *Y*-axis shows INF-γ^+^ T-cell count per 2.5 × 10^5^ lymphocytes. Significant groups were colored as red. Data are means ± SEM (standard error of the mean). Comparisons were performed by Student’s *t*-test (unpaired, two tailed). **** *p* < 0.0001; *** *p*-value < 0.001; ** *p*-value < 0.01; * *p*-value ≤ 0.05.

**Figure 4 vaccines-13-00239-f004:**
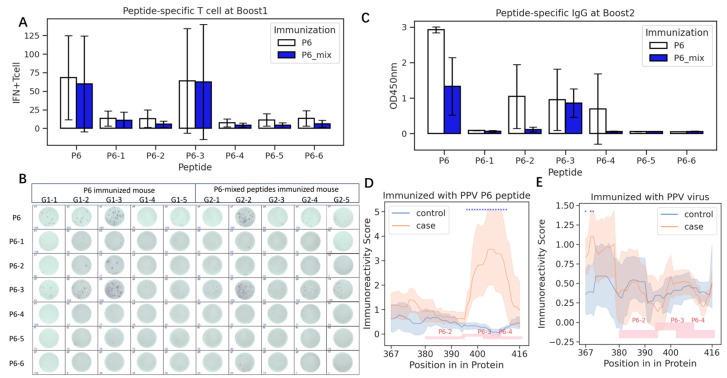
Precision of P6 peptide. (**A**,**B**) IFN-γ-secreting T-lymphocyte responses for P6 and P6-3 at Boost1. Each spot represent a single T cell secreting γ-IFN specific for peptides in rows. (**C**) Peptide-specific antibodies for P6 and P6-3 at Boost2. (**D**) Immunoreactivity scores along P6 peptides in P6 immunized mice. Red rectangles show the regions of P6-2, P6-3, and P6-4. Scores in positions with blue stars are significantly higher in case groups than control (*p* < 0.05). (**E**) Immunoreactivity scores along P6 peptides in PPV-immunized mice. Red rectangles show the regions of P6-2, P6-3, and P6-4. Scores in positions with blue stars are significantly higher in case groups than control (*p* < 0.05). Immunoreactivity score comparisons were performed by Mann–Whitney U test.

## Data Availability

The datasets used and/or analyzed during the current study are available from the corresponding author upon reasonable request.

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
