# Peer review of "The Identification of Dual T-Cell and B-Cell Epitopes Within Viral Proteins Utilizing a Comprehensive Peptide Array Approach"

_vaccines, 2025, doi:10.3390/vaccines13030239_

Round 1
Reviewer 1 Report
Comments and Suggestions for Authors
This work describes a methodology for identifying, selecting, and confirming dual T-cell and B-cell epitopes of viral protein with the potential cross-applicability to other pathogens and cancers. This is an interesting report worth accepting for publication in Vaccines. I only have a few comments
Comments:
01. It would be beneficial to validate the selected PPV capsid 1 14 peptides' immune reactivity with PPV-injected mouse serum as a supplementary experiment.
02. Please mention cells that are used to propagate the PPVA30 virus strain.
03. As mentioned in lines 72 and 73, Spleen tissues have been collected from PPV-injected mice, but there are no results in the manuscript. Please check this statement for accuracy.
04. Check Figure 3C y-axis title and values for accuracy
05. According to Figure 3E blood samples have been collected from 10 mice this shows a discrepancy with Figure 3A's depiction and explanation.
06. Check the Figure citations in the text for accuracy
Author Response
Point-by-point response to reviewer 1
We would like to thank the editor and the reviewers for their effort and time put into the review of the manuscript. Each comment has been carefully considered and addressed. The reviewers’ comments are indicated in red color. Responses to the reviewers are indicated in black color. Major revisions to the manuscripts are highlighted in yellow in the marked-up version of the revised manuscript.
Reviewer 1.
This work describes a methodology for identifying, selecting, and confirming dual T-cell and B-cell epitopes of viral protein with the potential cross-applicability to other pathogens and cancers. This is an interesting report worth accepting for publication in Vaccines. I only have a few comments
Reply:
We are grateful for your thoughtful evaluation of our manuscript and appreciate your positive assessment of our work. We are pleased that you find our methodology for identifying, selecting, and confirming dual T-cell and B-cell epitopes of viral proteins to be interesting and worthy of publication in Vaccines.
We acknowledge your kind comments and are eager to address the few points you have raised. Please find our responses below.
Comments:
- It would be beneficial to validate the selected PPV capsid 1 14 peptides' immune reactivity with PPV-injected mouse serum as a supplementary experiment.
Reply:
We appreciate your suggestion to validate the immune reactivity of the selected 14 PPV capsid peptides using PPV-injected mouse serum as a supplementary experiment. We agree that assessing the immune reactivity of these peptides would be beneficial in understanding the epitope characteristics of the target virus. This step is indeed an optional component of our vaccine peptide discovery process, considering the trade-offs between the benefits of additional validation and the constraints imposed by larger target proteins and project budgets.
In the PPV project, after identifying the 14 peptides, our priority was to evaluate the antibody titers. Due to the high sample consumption during the initial antibody titer experiments, which exceeded our expectations, we encountered difficulties in replenishing serum samples for further experimentation. Given the constraints of animal ethics and the significance of experimental validation, we decided not to pursue additional immune reactivity assessments for peptides that elicited relatively weaker antibody responses. We focused on evaluating the immune reactivity of the strongest peptide, P6, which was identified through T-cell and antibody experiments. While this decision may have resulted in a lack of comprehensive evaluation of all 14 peptides, we believe it was necessary to balance the experimental goals with the available resources and ethical considerations.
- Please mention cells that are used to propagate the PPVA30 virus strain.
Reply:
Thank you for bringing to our attention the need to declare the cell line used in our study. We had indeed explored several options, including Vero, MA104, and Swine testicular (ST) cell lines, before selecting the ST cell line as the most suitable for propagating the PPV30 virus strain. This information has now been added to the Methods section to provide clarity on our experimental approach.
- As mentioned in lines 72 and 73, Spleen tissues have been collected from PPV-injected mice, but there are no results in the manuscript. Please check this statement for accuracy.
Reply:
Regarding your comment about the collection of spleen tissues from PPV-injected mice, here we would like to clarify that these tissues were indeed collected for the purpose of conducting ELISpot assays. The T-cell studies presented in the manuscript, which are based on ELISpot assay results, were performed using these spleen tissue samples.
- Check Figure 3C y-axis title and values for accuracy
- According to Figure 3E blood samples have been collected from 10 mice this shows a discrepancy with Figure 3A's depiction and explanation.
Reply:
Thank you for your kindly reminding on the confusion results in Figure 3. We have corrected the y title in Figure 3B. In Figure 3C, because the OD450nm value was detected by peptide-ELISA, so total OD450nm values were calculated by summing up individual peptide values, which is only used to demonstrate an increasing trend with successive boosts. In Figure 3E, 3F and 3G, the X-axis shows the values of each peptide. We have added this explanation in main text.
- Check the Figure citations in the text for accuracy
Reply:
Thank you for your reminding to check the figure citations. There are really some mistakes, and they have been fixed in the updated version.
We appreciate your feedback and hope that these analyses will help you better understand our research results. If you have any further questions or suggestions, we would be delighted to hear them. Once again, thank you for your support and encouragement.

Reviewer 2 Report
Comments and Suggestions for Authors
Thank you for giving me the opportunity to review your valuable paper.
Peptide arrays are an excellent tool for epitope mapping of antigens recognized by antibodies and have been used in a variety of applications in recent years.
This study is a preliminary investigation of the application of Peptide Array technology to the creation of optimal epitope vaccines. This is a novel and interesting challenge.
In this study, it is likely that the goal is to seek epitope peptides that are recognized by both T and B cells, with the aim of creating effective epitope vaccines.
Whether using a peptide epitope that is commonly recognized by T-cell and B-cell receptors is superior to using a pathogen or some of its larger/full-length proteins as a vaccine must be discussed. I believe that data and explanations are needed to answer this question. Antibodies that are effective in preventing infection, and antibodies with high neutralizing activity and antibodies with high specificity and reactivity, do not necessarily match.
In the INTRODUCTION section, I believe it is necessary to explain the basic validity of the logic of this study in the light of basic concepts of immunology.
In searching for common peptide epitopes, there is the issue of binding affinity to the MHC.
In this study, authors were able to use the affinity of H-2-Db from the database, but in actual clinical use, it is necessary to deal with different and various MHCs. When peptide is used as an immunogen, the problem of MHC binding affinity cannot be ignored. I think that consideration and discussion on whether and how to resolve this issue is necessary as a future prospective.
I think that p1-14 is part of the structural protein of PPV, but an explanation regarding it is needed. The sentence “ Given that the length of the P6 peptide is up to 50 amino acids (line 230)” does not make sense to me well. It is described that “Each peptide includes 5-13 AA in length (line 105)” in the peptide array used in this study.
I think it is necessary to specify the amino acid sequence of each peptide(p1-p14) and P-6 sub-peptide (P6-1, P6-2, P6-3, P6-4, P6-5, P6-6) used.
Also, the position of p6-3 peptide, the most likely immunogen found in this study, in PPV needs to be explained.
I think it is necessary to specify the amino acid sequence of each peptide(p1-p14) and P-6 sub-peptide used.
Figure A1 (line 175) ➡ Figure 1 ?
Figure A2 (line 175) ➡ Figure 2 ?
Figure 1 (line 184) ➡ Figure 2 ?
Author Response
Point-by-point response to reviewer 2
We would like to thank the editor and the reviewers for their effort and time put into the review of the manuscript. Each comment has been carefully considered and addressed. The reviewers’ comments are indicated in red color. Responses to the reviewers are indicated in black color. Major revisions to the manuscripts are highlighted in yellow in the marked-up version of the revised manuscript.
Reviewer 2.
Thank you for giving me the opportunity to review your valuable paper.
Peptide arrays are an excellent tool for epitope mapping of antigens recognized by antibodies and have been used in a variety of applications in recent years.
This study is a preliminary investigation of the application of Peptide Array technology to the creation of optimal epitope vaccines. This is a novel and interesting challenge.
Reply:
We are thrilled to have the opportunity to engage with a professional who has been following the advancements in peptide microarray technology. The application of this technology in vaccine development is a highly meaningful and challenging field, particularly in the context of the COVID-19 pandemic. Since 2020, our team has been involved in various vaccine projects, including SARS-CoV-2 [unpublished], PPV, and MonkeyPox[1]. Although we have faced numerous setbacks, the integration of microarray data with deep learning algorithms has enabled us to develop an immune reactivity assessment model that can evaluate linear and spatial reactivity in monoclonal and polyclonal antibodies [unpublished] or serum samples [this manuscript, [1]].
Through this journey, we have gained a deeper understanding of the potential applications of peptide microarrays in this field. We aim to publish our findings in a series of articles, showcasing the progress we have made over the years, with the hope of contributing positively to the development of vaccine research. This manuscript, which focuses on the linear reactivity evaluation of peptide immunization, is the most accessible and easiest to understand for experts in the biomedical field. We hope to publish it in a reputable journal in this field and receive guidance from professionals like yourself.
We appreciate your expertise and look forward to your feedback, which will be invaluable in helping us improve our work. Thank you for considering our manuscript.
In this study, it is likely that the goal is to seek epitope peptides that are recognized by both T and B cells, with the aim of creating effective epitope vaccines.
Whether using a peptide epitope that is commonly recognized by T-cell and B-cell receptors is superior to using a pathogen or some of its larger/full-length proteins as a vaccine must be discussed. I believe that data and explanations are needed to answer this question.
Reply:
The primary objective of this research is indeed to identify linear epitopes that are recognized by both T and B cells. We aim to discover a peptide that can elicit a robust response from both T and B cells. But SARS-CoV-2 mutation studies suggests that the most effective peptides are often located in protein regions that are early targets for immune escape[2]. Therefore, we do not expect to necessarily find such a peptide or guarantee its suitability as a vaccine.
However, this research has potential value in the field of vaccinology in two key areas:
- For an unknown virus, identifying the most effective epitope peptides (those that can stimulate both T and B cells) is a crucial first step, whether for protecting these peptides to delay immune escape or using them as vaccine candidates.
- While current tumor vaccines mainly focus on T-cell epitopes[3], the evaluation of B-cell reactivity has received relatively little attention[4,5]. Nevertheless, we anticipate that this aspect will receive increasing attention in the future.
Based on these considerations, we believe that exploring the most effective peptides is a worthwhile approach. Fortunately, our efforts led to the identification of such a peptide in PPV.
Antibodies that are effective in preventing infection, and antibodies with high neutralizing activity and antibodies with high specificity and reactivity, do not necessarily match.
Reply:
We completely agree with your point that antibodies effective in preventing infection, those with high neutralizing activity, and those with high specificity and reactivity do not necessarily overlap. Any antibody that can bind strongly and specifically to an antigen protein of an infectious virus should have biological value, even if the exact nature and quantification of this value are not yet fully understood. Here, we believe that advances in technology will drive a deeper understanding of the complex relationships between antibody characteristics and their functional roles in preventing infection.
In the INTRODUCTION section, I believe it is necessary to explain the basic validity of the logic of this study in the light of basic concepts of immunology.
Reply:
We have reorganized the Introduction section to provide a clearer explanation of the study's logic and significance. We have made an effort to present the underlying rationale and immunological principles that guide our research, with the aim of providing a more comprehensive and accessible introduction to the study.
In searching for common peptide epitopes, there is the issue of binding affinity to the MHC.
In this study, authors were able to use the affinity of H-2-Db from the database, but in actual clinical use, it is necessary to deal with different and various MHCs. When peptide is used as an immunogen, the problem of MHC binding affinity cannot be ignored. I think that consideration and discussion on whether and how to resolve this issue is necessary as a future prospective.
Reply:
We appreciate your emphasis on the crucial issue of MHC binding affinity in the development and application of peptide-based vaccines. While we briefly discussed this topic in the Discussion section, here we would like to provide a more detailed explanation of our understanding of the long-term solutions to this challenge.
Firstly, we believe that individual MHC typing will become increasingly accessible and affordable in the future, potentially through low-cost detection schemes. Our laboratory is currently exploring the use of chip technology to achieve breakthroughs in this area. With the aid of next-generation sequencing technologies, chip technologies, and advanced artificial intelligence algorithms, we anticipate that MHC typing will become as routine as blood type testing. In fact, cancer patients are already benefiting from MHC typing through sequencing technologies.
Secondly, it is well established that different MHC alleles exhibit varying frequencies in populations[6] and confer distinct susceptibilities to infectious diseases[7,8]. This suggests that future vaccines could be tailored to specific MHC types, particularly those with high susceptibility. Additionally, the development of mixed-peptide vaccines will be an important direction for infectious disease vaccines, as combining different peptides can enhance vaccine applicability across diverse populations. For cancer vaccines, multi-antigen targeting through mixed-peptide approaches may also effectively improve vaccine efficacy.
In summary, we acknowledge the significance of MHC binding affinity in peptide-based vaccine development and are committed to addressing this challenge through innovative solutions, including advances in MHC typing, mixed-peptide vaccine design, and personalized medicine approaches.
I think that p1-14 is part of the structural protein of PPV, but an explanation regarding it is needed. The sentence “ Given that the length of the P6 peptide is up to 50 amino acids (line 230)” does not make sense to me well. It is described that “Each peptide includes 5-13 AA in length (line 105)” in the peptide array used in this study.
I think it is necessary to specify the amino acid sequence of each peptide(p1-p14) and P-6 sub-peptide (P6-1, P6-2, P6-3, P6-4, P6-5, P6-6) used.
Also, the position of p6-3 peptide, the most likely immunogen found in this study, in PPV needs to be explained.
I think it is necessary to specify the amino acid sequence of each peptide(p1-p14) and P-6 sub-peptide used.
Reply:
Firstly, we appreciate your reminder about making the sequences publicly available. We have prepared Supplementary Table 1, which contains the sequences of all peptides mentioned in the article. This will facilitate readers' understanding of our research results.
Secondly, we attempted to explain the efficacy of p6-3. We conducted two analyses: one on the antibody reactivity characteristics of mice, and the other on the classical vaccine peptide screening. The results have been shown in Figure S5. Through these analyses, we believe that the efficacy of p6-3 can be explained.
In the first analysis, we found that mice immunized with P6 showed relatively strong reactivity to specific space-kmers like VP.HL and P.HL.R contained in p6-3. This suggests that the presence of these space-kmers in p6-3 is one reason for its efficacy. However, we must note that not all peptides containing these space-kmers will necessarily have strong reactivity, as our deep learning model considers the entire peptide sequence, not just these space-kmers.
In the second analysis, we used the Kolaskar-Tongaonkar method from IEDB to evaluate the immunogenicity of the Capsid1 protein sequence. The results showed that the p6-3 sequence had high scores, particularly at P402, which is consistent with our previous findings of the main binding motifs VP.HL and P.HL.R. This provides explanation for the efficacy of p6-3.
Figure S5. Possible important residuals explaining the efficiency of P6-3. A. With the high-signal peptide probes of each sample, we calculated the Zscore to define the enrichement of space-kmers by comparing to a background distribution of them. Then we also used the immunoreactivity model to predict reactivity of each space-kmer. All the enriched space-kmers were shown in scatterplot. Space-kmers with reactivity >=3 and included in P6-3 peptide or P6-4 peptide were colored red and blue respectively. B. Antigenicity was evaluated with Kolaskar-Tongaonkar method from IEDB. The P6-3 region formed a high peak and was colored red.
Figure A1 (line 175) ➡ Figure 1 ?
Figure A2 (line 175) ➡ Figure 2 ?
Figure 1 (line 184) ➡ Figure 2 ?
Reply:
We would like to express our sincere gratitude for pointing out the error in the figure numbering. We have made the necessary corrections and thoroughly reviewed the numbering of all subsequent figures to ensure accuracy.
We appreciate your feedback and hope that these analyses will help you better understand our research results. If you have any further questions or suggestions, we would be delighted to hear them. Once again, thank you for your support and encouragement.
Reference:
- Zai, X. A Deglycosylated Subunit Vaccine Provides Complete Protection Against Lethal Orthopoxvirus Challenge. Science translational medicine 2025.
- Carabelli, A.M.; Peacock, T.P.; Thorne, L.G.; Harvey, W.T.; Hughes, J.; COVID-19 Genomics UK Consortium; De Silva, T.I.; Peacock, S.J.; Barclay, W.S.; De Silva, T.I.; et al. SARS-CoV-2 Variant Biology: Immune Escape, Transmission and Fitness. Nat Rev Microbiol 2023, doi:10.1038/s41579-022-00841-7.
- Wang, S.H.; Cao, Z.; Farazuddin, M.; Chen, J.; Janczak, K.W.; Tang, S.; Cannon, J.; Baker, J.R. A Novel Intranasal Peptide Vaccine Inhibits Non-Small Cell Lung Cancer with KRAS Mutation. Cancer Gene Ther 2024, doi:10.1038/s41417-023-00717-9.
- Lynch, K.T.; Squeo, G.C.; Kane, W.J.; Meneveau, M.O.; Petroni, G.; Olson, W.C.; Chianese‐Bullock, K.A.; Slingluff, C.L.; Foley, E.F.; Friel, C.M. A Pilot Trial of Vaccination with Carcinoembryonic antigen and Her2/Neu Peptides in Advanced Colorectal Cancer. Intl Journal of Cancer2022, 150, 164–173, doi:10.1002/ijc.33793.
- Ma, J.; Wu, Y.; Ma, L.; Yang, X.; Zhang, T.; Song, G.; Li, T.; Gao, K.; Shen, X.; Lin, J.; et al. A Blueprint for Tumor-Infiltrating B Cells across Human Cancers. Science 2024, 384, eadj4857, doi:10.1126/science.adj4857.
- Zhou, F.; Cao, H.; Zuo, X.; Zhang, T.; Zhang, X.; Liu, X.; Xu, R.; Chen, G.; Zhang, Y.; Zheng, X.; et al. Deep Sequencing of the MHC Region in the Chinese Population Contributes to Studies of Complex Disease. Nat Genet 2016, 48, 740–746, doi:10.1038/ng.3576.
- Saif, N.; Griffin, G.K. The HLA Advantage: Why COVID-19 Benched You but Not Your Co-Worker. Sci. Immunol. 2023, 8, eadk5067, doi:10.1126/sciimmunol.adk5067.
- Fricke-Galindo, I.; Falfán-Valencia, R. Genetics Insight for COVID-19 Susceptibility and Severity: A Review. Front. Immunol. 2021, 12, 622176, doi:10.3389/fimmu.2021.622176.

Round 2
Reviewer 2 Report
Comments and Suggestions for Authors
Thank you for giving me the opportunity to review your valuable paper.
I thank you for your thoughtful response to my doubts and questions.
I believe you have answered them, including new materials and references.
I also believe that you have mentioned the issues and problems with the research that I have pointed out here, including in your discussion.
I am satisfied with your revisesd version.